# Isolation and Functional Characterization of a *LEAFY* Gene in Mango (*Mangifera indica* L.)

**DOI:** 10.3390/ijms23073974

**Published:** 2022-04-02

**Authors:** Yihan Wang, Haixia Yu, Xinhua He, Tingting Lu, Xing Huang, Cong Luo

**Affiliations:** State Key Laboratory for the Conservation and Utilization of Subtropical Agro-Bioresources, College of Agriculture, Guangxi University, 100 East Daxue Road, Nanning 530004, China; wyhslb@163.com (Y.W.); yuhaixia0201@163.com (H.Y.); lutting4040@163.com (T.L.); xinghuangkeys@163.com (X.H.)

**Keywords:** mango (*Mangifera indica* L.), *LEAFY*, expression analysis, functional identification, promoter analysis

## Abstract

*LEAFY* (*LFY*) plays an important role in the flowering process of plants, controlling flowering time and mediating floral meristem differentiation. Owing to its considerable importance, the mango *LFY* gene (*MiLFY*; GenBank accession no. HQ585988) was isolated, and its expression pattern and function were characterized in the present study. The cDNA sequence of *MiLFY* was 1152 bp, and it encoded a 383 amino acid protein. *MiLFY* was expressed in all tested tissues and was highly expressed in flowers and buds. Temporal expression analysis showed that *MiLFY* expression was correlated with floral development stage, and two relative expression peaks were detected in the early stages of floral transition and floral organ differentiation. Moreover, 35S::GFP-MiLFY fusion protein was shown to be localized to the nucleus of cells. Overexpression of *MiLFY* in *Arabidopsis* promoted early flowering and the conversion of lateral meristems into terminal flowers. In addition, transgenic plants exhibited obvious morphological changes, such as differences in cauline leaf shape, and the number of lateral branches. When driven by the *MiLFY* promoter, *GFP* was highly expressed in leaves, floral organs, stems, and roots, during the flowering period. Exogenous gibberellin (GA_3_) treatment downregulated *MiLFY* promoter expression, but paclobutrazol (PPP_333_) upregulated it. Bimolecular fluorescence complementation (BiFC) assays showed that the MiLFY protein can interact with zinc-finger protein 4 (ZFP4) and SUPPRESSOR OF OVEREXPRESSION OF CONSTANS 1 (MiSOC1D). Taken together, these results indicate that *MiLFY* plays a pivotal role in controlling mango flowering, and that it is regulated by gibberellin and paclobutrazol.

## 1. Introduction

Flowering is an important process in the plant life cycle, and the time of flowering (early or late) essentially determines the time of harvest. In recent decades, much progress has been made in understanding the physiological and molecular mechanisms underlying flowering time in plants. It is clear that plant flowering is mainly affected by various environmental and endogenous signals, such as daylength, temperature, drought, plant age, endogenous phytohormones, and exogenous plant growth regulators (PGRs) [1,2]. A complex gene regulatory network has been revealed in *Arabidopsis.* At least six flowering pathways and several genes, such as *FLOWERING LOCUS T* (*FT*), *SUPPRESSOR OF OVEREXPRESSION OF CONSTANS 1* (*SOC1*), and *LEAFY* (*LFY*), which act as floral pathway integrators to activate downstream floral meristem identity genes, such as *LFY* and *APETALA1* (*AP1*), cooperate to promote flowering [3].

*LFY* is a plant-specific transcription factor that is a master regulator of flower initiation and, as such, it determines floral fate in *Arabidopsis* [4]. Many plant genomes contain single copies or low copy numbers of *LFY* homologs, and these genes contain three exons and two introns at conserved positions [5]. The *LFY* gene is highly conserved across plant species. LFY proteins contain conserved C-terminal and N-terminal regions, which function in the regulation of transcriptional activity and have a conserved DNA-binding domain [6,7]. Overexpression of *LFY* in transgenic *Arabidopsis* and tobacco can induce early flowering [8,9]. Plants with a mutant *LFY* gene develop leaves and associated shoots instead of flowers [4]. Overexpression of *AtLFY* in citrus trees resulted in precocious flowering phenotypes, with flowers developing during the juvenile period [10]. However, overexpression of the tobacco *LFY* homolog *NFL1* in *Arabidopsis* did not severely affect flowering [11]. Therefore, further study of *LFY* homologs in different species is needed to improve our understanding of the *LFY* gene function in flowering regulation.

Mango (*Mangifera indica* L.) is one of the most important woody tree species, and it is widely distributed in tropical and subtropical regions. Mango has a long juvenile period, and its flowering is triggered by cold temperatures and rain. Identifying approaches for regulating the time of flowering to avoid adverse weather impacts is one of the most important research topics for mango production [1]. Understanding the molecular mechanisms underlying flowering regulation in mango can provide a theoretical basis for the regulation of flowering. To date, several flowering-related genes have been isolated and functionally characterized in mango [12,13,14,15]. However, functional information on the mango *LFY* gene (*MiLFY*) is not available. In the present study, we report an *LFY* gene from mango, and evaluate its expression in different tissues and at different times via quantitative real-time PCR (qRT–PCR). We explored the function of *MiLFY* by studying its heterologous expression in *Arabidopsis.* In addition, we developed a construct in which the expression of the *GUS* gene was driven by the *MiLFY* promoter to detect transcriptional activity at different developmental stages in transgenic *Arabidopsis*, and we analyzed the effects of gibberellin (GA_3_) and paclobutrazol (PPP_333_) treatment on its activity.

## 2. Results

### 2.1. Sequence and Phylogenetic Analysis of LFY

The sequence of a *MiLFY* homolog in mango was obtained. The cDNA length was 1152 bp, encoding a 383 amino acid protein (GenBank accession no. HQ585988). The full-length DNA sequence of *MiLFY* was 2170 bp, comprising three exons and two introns (Figure 1A). The predicted protein molecular weight was 43.31 kDa, and the isoelectric point was 6.37. The N- and C-terminal regions of the MiLFY proteins were highly conserved across species (Figure 1B). The C- and N-terminal regions contained a DNA-binding domain and a sterile alpha motif (SAM) domain. The MiLFY protein had 61.79% sequence homology with the *Arabidopsis thaliana* ortholog. Phylogenetic tree analysis showed that MiLFY was closely related to DlLFY (*Dimocarpus longan*), CsLFY (*Citrus sinensis*), and ClLFY (*Clausena lansium*) (Figure 1C).

### 2.2. Analysis of MiLFY Gene Expression in Mango

The expression patterns of the *MiLFY* gene were analyzed via qRT–PCR. As shown in Figure 2A, *MiLFY* was expressed in all tested tissues, albeit at different levels. The expression of *MiLFY* was higher in the tissues of flowering branches than in those of nonflowering branches. The expression level was highest in flowers and lowest in leaves. The expression patterns of *MiLFY* in mature stems, mature leaves, and buds, at different flower development stages (Figure 2B–D), were determined by qRT–PCR. *MiLFY* was more highly expressed in buds than in mature leaves. The highest expression level was found during the floral organ differentiation stage (January 2019) in buds, and during the flowering transition stage in leaves. However, *MiLFY* was highly expressed in mature stems at the flowering stage and flowering transition stage in the TaiNong No. 1 cultivar, but was highly expressed at the floral organ differentiation stage in the ‘SiJiMi’ cultivar.

### 2.3. Subcellular Localization of MiLFY

To determine the subcellular localization of MiLFY, the protein was fused to green fluorescent protein (GFP) under the control of the constitutive cauliflower mosaic virus 35S (CaMV35S) promoter. The 35S::GFP-MiLFY vector and 35S::GFP-P1300 vector (control) were introduced into onion epidermal cells via *Agrobacterium*. As shown in Figure 3, the fluorescence signal of the 35S::GFP-P1300 control vector was observed throughout the cell. However, 35S::GFP-MiLFY fusion proteins were only visible in the nucleus, and were colocalized with 4′,6-diamidino-2-phenylindole (DAPI).

### 2.4. Phenotypic Analysis of Transgenic Plants with Overexpression

A vector overexpressing the *MiLFY* gene was transformed into wild-type (WT) *Arabidopsis* to study the function of *MiLFY*. WT plants were used as negative controls, and WT plants transformed with the pBI121 empty vector were used as positive controls. Semiquantitative PCR was used to detect the expression level of the exogenous *MiLFY* gene in transgenic plants, and qRT–PCR was used to detect the expression level of the *A**rabidopsis* flowering-related genes *AtFT*, *AtAP1*, and *AtSOC1* in transgenic and control plants.

*MiLFY* was expressed normally in transgenic *Arabidopsis thaliana*, but not in the control plants (Figure 4A-a1). Compared with the control plants, the transgenic plants bolted and flowered earlier (Figure 4A-a,B-a). The flowering time ranged from 24.42 to 27.38 d in the transgenic plants, and from 31.3 to 31.5 d in the control plants when the plants were cultivated under long-day conditions (Figure 4B-b). The average number of rosette leaves in the transgenic plants was between 5.89 and 6.64, which was lower in the control plants (between 8.3 and 8.4) (Figure 4B-c). All the transgenic plants with overexpression, except for the OE-13 plants, were shorter than the control plants. The shortest line was OE-31, at 14.86 ± 0.67 cm, which was approximately 11 cm shorter than the control lines (Figure 4B-d).

Transgenic plants overexpressing the *MiLFY* gene exhibited different morphological phenotypes (Figure 4A). All axillary rosette branches (basal branches) and main stem branches (stem branches) developed single flowers (Figure 4A-b,c,f). In addition, the pedicels were wrapped by curly cauline leaves, and the parts of the single flowers were opposite or whorled on the main stem (Figure 4A-c,d). Unusual siliques developed on the upper stems and basal branches (Figure 4A-e,g,i), and the transgenic plants had fewer flower petals than the control plants (Figure 4A-h,j). All transgenic *MiLFY*1-overexpressing *Arabidopsis* lines except OE-13 had the same morphological characteristics. The morphology of the OE-13 line was similar to that of the control lines.

The experimental materials were sampled when the transgenic plants were flowering and the control plants were not. RNA was extracted from the aerial portions of the plants 27 d after germination, and this RNA was subsequently reverse-transcribed into cDNA. qRT–PCR was used to determine whether the expression of endogenous genes in transgenic *Arabidopsis* was affected by *MiLFY* overexpression (Figure 5). Compared with the WT *Arabidopsis* plants, and the plants transformed with the empty vector, the transgenic *MiLFY*1-overexpressing *Arabidopsis* plants exhibited obviously increased expression of *At**AP1*, *At**FT*, and *At**SOC1*.

### 2.5. Activity of GUS Driven by the MiLFY Promoter in Transgenic Arabidopsis Lines

In our previous study, the *Mi**LFY* promoter was cloned, and its cis-elements were analyzed [16]. In the present study, the 35S promoter in the pBI121 vector was replaced with the *MiLFY* promoter to drive *GUS* gene expression. The pLFY-GUS vector was subsequently transformed into WT *Arabidopsis*, and WT plants and pBI121-GUS vector-transformed plants were used as the controls. Different transgenic and control seedlings, as well as organs of mature plants, were subjected to histochemical staining for GUS to detect *MiLFY* promoter expression. As shown in Figure 6, the stem and stem apex in the transgenic *Arabidopsis* plants exhibited sites of *GUS* expression at 5 d (two-true leaf stage) and 10 d (four-true leaf stage). Moreover, *GUS* expression was detected in all mature organs, including flowers, stems, leaves, siliques, and roots. The expression level of the *MiLFY* promoter was relatively high in floral organs and stems. However, no staining was detected in control samples.

Many PGR response elements have been found in the *MiLFY* promoter [17]. In this study, 15 d old transgenic plants harboring the *MiLFY* promoter were treated with GA_3_ and PPP_333_, and the control lines were treated with ddH_2_O. Promoter activity was affected by the PGRs, as determined by analyzing the expression level of the *GUS* gene. Compared with those in the plants treated with ddH_2_O, the *GUS* expression levels in the transgenic plants harboring the *MiLFY* promoter were reduced by GA_3_ treatment. In contrast, the *GUS* expression level was significantly increased by PPP_333_ treatment (Figure 7).

### 2.6. Yeast Two-Hybrid (Y2H) Screening and Confirmation by Bimolecular Fluorescence Complementation (BiFC) Assays

Using Y2H screens, we identified the following two candidate-interacting proteins: zinc-finger protein 4 (ZFP4) and MiSOC1D. Further BiFC assays using a split yellow fluorescent protein (YFP) system in onion epidermal cells were used to verify the interactions. The coexpression of MiLFY and MiZFP4, and of MiLFY and MiSOC1D, resulted in YFP signals in the nucleus of onion epidermal cells (Figure 8).

## 3. Discussion

*LFY* not only regulates flowering time but also has a specific function as a floral meristem identity gene in the flower development pathway [18]. Moreover, *LFY* links floral induction with flower initiation [19]. In the present study, a *LFY* homolog from the mango cultivar SiJiMi was identified and named *MiLFY*. The function of the *MiLFY* gene and the regulation of its promoter activity were systematically analyzed, thereby helping to determine the genetic and molecular mechanisms underlying the involvement of *LFY* genes in mango flowering.

Many woody fruit tree species, such as navel orange [6], grapevine [20], *Ziziphus jujube* [21], and longan [22], contain only a single copy of the *LFY* gene. However, multiple copies are present in various species, namely, pear [23], Chinese quince [23], and loquat [24], all of which contain two copies. Only a single copy of *MiLFY* was found in mango. Phylogenetic analysis indicated that *MiLFY* is more closely related to its orthologs in longan, citrus, and wampee than to those in other woody fruit tree species. The N- and C-terminal regions of the LFY protein are highly conserved across species; this conserved structure assures the similarity of *LFY* gene function [25].

The expression pattern of the *LFY* gene varies among woody species. For example, the London planetree *PlacLFY* gene was found to be expressed mainly in male and female inflorescences and was only weakly expressed in stems and young leaves [26]. Precocious trifoliate orange *CiLFY* is highly expressed in mature apex bud, flower, and stem tissues, but not in juvenile tissues; high *CiLFY* expression was found to be maintained from December to March [27]. *Jatropha curcas JcLFY* is expressed in inflorescence buds, flower buds, and carpels, with the highest expression occurring in the early developmental stage of flower buds [28]. *Prunus mume PmLFY* is highly expressed in floral buds, leaf buds, pistils, and seeds, with the highest expression occurring in floral buds during the floral differentiation stage [7]. However, peach PpLFL was expressed mainly in leaves and in the petal primordia of the shoot apical meristem during the floral induction period [29]. *Populus tomentosa PtLFY* mRNA was found to be highly abundant in the roots and floral buds of both male and female flowers [30]. In the present study, *MiLFY* was more highly expressed in the flowers and stems of flowering branches than in those of nonflowering branches, and its expression was correlated with the floral development stage in different organs in different cultivars. Litchi *LcLFY* was expressed primarily in flower buds, but was barely detectable in stems, mature leaves, petioles, and pedicels [31].

Overexpression of the *LFY* gene causes early flowering in many plant species, although the overall plant morphology remains normal [7,29,31,32]. However, some differences do occur. *JcLFY* overexpression induces early flowering but causes the production of single flowers and terminal flowers in *Arabidopsis* [28]. Overexpression of *VpLFY2* without overexpression of *VpLFY1* causes precocious flowering in *Arabidopsis* [31]. *AfLFY* expression in transgenic tobacco plants promotes precocious flowering, and these transgenic plants exhibit obvious changes in leaf shape [33]. In the present study, overexpression of *MiLFY* promoted early flowering in *Arabidopsis*, but considerable morphological variation was observed. For example, indeterminant inflorescences became single flowers, and cauline leaves were curled and intertwined with pedicels. In addition, the number of flower petals was decreased in the transgenic plants, and some pods were short or curved. The phenotype of the transgenic plants was similar to that of plants overexpressing *PmLFY1* [7]. These results suggest that the *LFY* gene can promote flower formation in different species but differentially influences plant development.

Gene expression is regulated by cis-elements in promoters. *PlacLFY* promoter activity was detected in the shoot apices, young leaves, young fruits, petioles, and young/old stems of pPlacLFY::GUS transgenic tobacco, consistent with the expression pattern of *PlacLFY* in London planetree [26]. *CcLFY* promoter expression is influenced by low temperature and dark conditions [34]. The promoter of *MiLFY* in mango was previously characterized [17]. In the present study, the *MiLFY* gene promoter was transformed into *Arabidopsis*, and analysis of GUS staining showed that the *MiLFY* promoter was active mainly in flowers, stems, leaves, and roots during the flowering period, consistent with the expression patterns of *MiLFY* in mango (Figure 2). Exogenous GA_3_ promotes *Arabidopsis* flowering but inhibits mango flowering, and PPP_333_, a synthetic inhibitor of GA_3_, promotes mango flower formation [1]. In the present study, we treated *MiLFY* promoter-containing transgenic seedlings with exogenous GA_3_ and PPP_333_ and found that the expression of the *MiLFY* promoter was significantly inhibited by GA_3_ but enhanced by PPP_333_. These results suggest that the *MiLFY* gene is involved in phytohormone-mediated regulation of mango flowering.

As a floral integration factor, the *LFY* gene plays an important regulatory role in the floral network [35]. Winter (2011) identified direct *LFY* target genes throughout the genome. These target genes were found to be involved in flowering time, floral organ development, phytohormone responses, and biotic stimulus responses. In the present study, we found that two proteins, MiZFP4 and MiSOC1D, can directly interact with LFY. ZFPs constitute one of the largest transcription factor families, whose members are highly involved in transcriptional regulation of flowering induction, floral organ morphogenesis, and stress responses [36]. *SOC1* is a floral integration factor, and overexpression of *MiSOC1* promotes early flowering in *A. thaliana* [37].

In conclusion, we functionally characterized the *MiLFY* gene in mango and found that overexpression of *MiLFY* significantly promoted flowering in transgenic plants. Moreover, transgene expression led to significant morphological variation, including changes in floral organs and leaf morphology, in the transgenic plants. For the first time, we found that GA_3_ treatment inhibits, but PPP_333_ promotes, *MiLFY* promoter activity. This pattern is consistent with the finding that GA_3_ inhibits mango flower development while PPP_333_ promotes mango flowering. Protein interaction analysis showed that by directly interacting with MiZFP4 and MiSOC1D, the MiLFY protein regulates mango flowering.

## 4. Materials and Methods

### 4.1. Plant Materials

The mango cultivars SiJiMi and TaiNong No. 1 were grown in an orchard of the College of Agriculture, Guangxi University, Nanning, Guangxi, China. For tissue expression analysis, leaves, stems, and buds or flower tissues were collected from the flowering and nonflowering branches of the same plants on 4 January 2019. For seasonal expression analysis, leaves, stems, and buds or flowers were collected each month from 5 November 2018 to 6 March 2019. All samples were used for experiments before storage at −80 °C. The *Arabidopsis* ecotype Columbia (Col-0) plants used for transformation were maintained in our laboratory.

### 4.2. Isolation of the MiLFY Gene

Total RNA was extracted from mango leaves by using an RNAprep pure kit (DP441) (Tiangen, Beijing, China). First-strand cDNA was synthesized with M-MLV reverse transcriptase (TaKaRa, Dalian, China) using 1 µg of RNA according to the manufacturer’s instructions. Genomic DNA was isolated using the cetyl-trimethylammonium bromide (CTAB) method with minor modifications. In our previous study, we obtained the sequence of a *LFY* gene from transcriptome data from the mango cultivar SiJiMi (unpublished data) and, in this study, we further verified the correctness of its sequence by RT–PCR with the primers LFY-F and LFY-R (Appendix A). PCR amplification was performed as described in a previous study [2]. The primers LFY-F and LFY-R were also used to amplify genomic DNA.

### 4.3. Sequence Alignment and Bioinformatic Analysis

Sequence analysis and amino acid predictions were performed using BioXM 2.6 software (http://cbi.njau.edu.cn/BioXM/; 21 May 2021). The exon–intron structure was generated with IBS 1.0. The conserved protein domain was predicted with the NCBI Conserved Domain Database online tool (http://www.ncbi.nlm.nih.gov/Structure/cdd/cdd.shtml/; 11 May 2021). Multiple sequence alignment of the LFY protein was performed with DNAMAN software. A phylogenetic tree of LFY was constructed using the neighbor-joining method with MEGA 6.0, with 1000 bootstrap replicates (https://megasoftware.net/; 20 March 2022).

### 4.4. Analysis of MiLFY Gene Expression

The expression pattern of *MiLFY* was analyzed with an ABI 7500 Real-Time PCR System (Applied Biosystems, Foster City, CA, USA) and SYBR Premix Ex Taq II (Takara, Dalian, China) according to the manufacturers’ instructions. The reaction mixture contained the following components: reaction solution 10 μL, cDNA 2 µL (50 ng/μL), up- and downstream primers 0.5 μL (10 μM) each, ROX reference dye II 0.8 μL, and sterile water to 20 μL. PCR amplification was performed with the following thermal cycling program: 95 °C for 30 s; 40 cycles at 95 °C for 5 s, 60 °C for 34 s, and 95 °C for 15 s; melting curve analysis was performed at 95 °C for 15 s and 60 °C for 1 min. The gene-specific primers used are listed in Appendix A. *MiActin1* was used as the internal reference gene in mango [16]. The delta-delta Ct method was used to calculate the relative gene expression [38]. The data are presented as the average of at least three technical replicates.

### 4.5. Subcellular Localization Analysis

To analyze the subcellular localization of MiLFY, the full coding sequence (CDS) of *MiLFY* without the termination codon was inserted into the *Xba* I and *BamH* I sites in the P1300 vector to express the 35S::GFP-MiLFY fusion construct under the control of the CaMV35S promoter [39]. The 35S::GFP-MiLFY vector was transformed into *Agrobacterium*
*tumefaciens* EHA105a, and the transformants were used to infect onion (*Allium cepa*) epidermal cells. The onion epidermal cells were then observed at a wavelength of 488 nm with a confocal laser-scanning microscope (TCS-SP8MP, Leica, Germany). Nuclei were confirmed by DAPI staining.

### 4.6. Vector Construction and Transformation of Arabidopsis

The full CDS of the *MiLFY* gene was inserted into the *Xba*I and *Xma*I sites in pBI121 under the control of the CaMV35S promoter to construct the overexpression vector. The pBI121-MiLFY and pBI121 vectors were separately transformed into WT *Arabidopsis* via *A. tumefaciens* EHA105 using the floral dip method [40]. The transgenic plants were first grown on half-strength Murashige and Skoog (MS) medium supplemented with kanamycin (100 mg/L). The positive transgenic plants were further confirmed via PCR-based amplification of DNA. Homozygous T3 transgenic plants were used for subsequent experiments.

The bolting time, flowering time, number of rosette leaves, and plant height of the WT *Arabidopsis* and empty vector-transformed *Arabidopsis* plants (which were used as controls) were recorded or measured under long-day (LD) conditions. For semiquantitative PCR and qRT–PCR analysis, 24-d-old transgenic and WT *Arabidopsis* seedling leaves were collected for total RNA extraction. Total RNA was extracted, and first-strand cDNA was synthesized as described above. Semi-quantification was mainly used to detect whether *MiLFY* gene was normally expressed in transgenic plants. qRT–PCR was used to measure the expression levels of the endogenous flowering-related genes in *Arabidopsis*. The *Arabidopsis Actin*2 gene was used as the internal control. The semiquantitative PCR and qRT–PCR methods were described in a previous study [15].

### 4.7. MiLFY Promoter Activity Assay

In our previous study, a 1314-bp fragment of the *MiLFY* promoter was amplified through thermal asymmetric interlaced PCR (TAIL-PCR) [17]. This fragment was inserted into the pBI121 vector in place of the 35S promoter to activate GUS, and plants containing the 35S promoter vector and WT plants were used as the controls. pMiLFY-pBI121 transgenic plants were obtained via the above approach. Homozygous T3 transgenic plants were used for subsequent experiments. Growing plants at different stages were harvested and immersed in GUS staining buffer (Real-Time, Beijing, China) at the same time. The plant material was incubated in a constant-temperature incubator at 37 °C for 24 h in the dark and was then decolorized with 75% ethanol until the control samples became white. Images showing sites of GUS staining were acquired using an ultra-depth-of-field 3D microscopy system (VHX-6000, Leica, Germany).

To determine the effects of plant growth regulators (PGRs) on *MiLFY* promoter activity, 20-d-old *Arabidopsis* plants were sprayed with 150 mL of ddH_2_O, 10 μM GA_3_, or 10 μM PPP_333_. Materials were collected at 0 d, 3 d, and 6 d after treatment. Total RNA was extracted from these samples using an RNAprep pure kit (DP441) (Tiangen, Beijing, China), and the *GUS* gene expression level was measured via qRT–PCR. The gene-specific primers used are listed in Appendix A. The method was the same as that described above.

### 4.8. Y2H Screening and Confirmation by BiFC Assay

The full-length CDS of *MiLFY* was amplified and inserted into the pGBKT7 vector. A cDNA library comprising bud, leaf, and flower samples, obtained from SiJiMi mango trees during flower development, was constructed and stored in our laboratory. cDNA library screening was performed according to the Matchmaker Gold Yeast Two-Hybrid System User Manual (Clontech). All positive clones were sequenced and functionally annotated using the BLAST online search engine tool. To further verify protein–protein interactions, the full-length CDSs of the candidate-interacting proteins were inserted into the pSPYCE vector, and the full-length CDS of *MiLFY* was inserted into the pSPYNE vector. All fusion constructs were subsequently transformed into *A. tumefaciens* (strain GV3101). Subsequently, the different fusion vectors were transformed into onion epidermal cells. Fluorescence signals were observed 48 h after infiltration using laser-scanning confocal microscopy (TCS SP8, Leica, Germany)

### 4.9. Statistical Analysis

SPSS 19.0 statistical software (SPSS, Inc., Chicago, IL, USA) was used for statistical analysis.

## Figures and Tables

**Figure 1 ijms-23-03974-f001:**
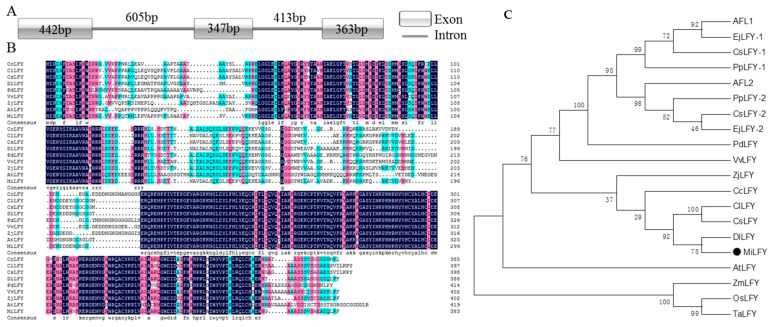
Sequence analysis of *LFY* genes. (**A**) Gene structure of *MiLFY* genes. (**B**) Amino acid sequence alignment of LFY proteins from different species. (**C**) Phylogenetic tree of LFY proteins from various species. The species, gene names, and GenBank accession numbers are as follows: *Citrus sinensis* (CsLFY, AY338976.1), *Carya cathayensis* (CcLFY, DQ989225.1), *Pyrus pyrifolia* (PpLFY-1, AB162029.1), *Pyrus pyrifolia* (PpLFY-2, AB162035.1), *Ziziphus jujuba* (ZjLFY, JN165097.2), *Dimocarpus longan* (DlLFY, EF489297.1), *Vitis vinifera* (VvLFY, XM_002284628.3), *Eriobotrya japonica* (EjLFY-1, AB162033.1), *Eriobotrya japonica* (EjLFY-2, AB162039.1), *Malus domestica* (AFL1, AB162028.1), *Malus domestica* (AFL2, AB056159.1), *Chaenomeles sinensis* (CsLFY-1, AB162032.1), *Chaenomeles sinensis* (CsLFY-2, AB162038.1), *Prunus dulcis* (PdLFY, AY947465.1), *Clausena lansium* (ClLFY, DQ497006.2), *Triticum aestivum* (TaLFY, BAE78663.1), *Oryza sativa* (OsLFY, AHX83809.1), and *Zea mays* (ZmLFY, ABC69153.1).

**Figure 2 ijms-23-03974-f002:**
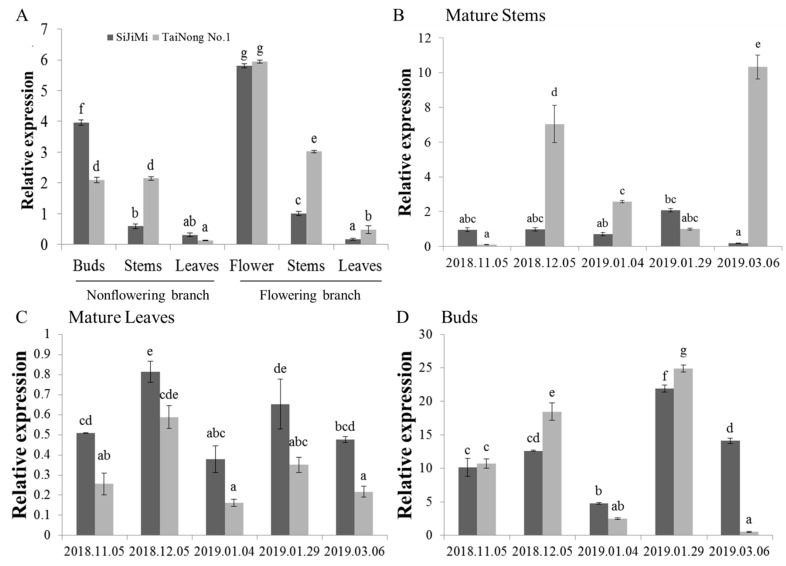
Relative expression levels of the *MiLFY* gene. (**A**) Tissue expression patterns of the *Mi**LFY* gene in buds/flowers, stems, and leaves of nonflowering and flowering branches. (**B**) Expression patterns of the *MiLFY* gene in mature stems during different flower development stages. (**C**) Temporal expression pattern of the *MiLFY* gene in mature leaves during different flower development stages. (**D**) Temporal expression pattern of the *MiLFY* gene in buds during different flower development stages. Significant differences among the samples were assessed at the *p* < 0.01 level by Student’s t tests; different letters indicate significance between different samples.

**Figure 3 ijms-23-03974-f003:**
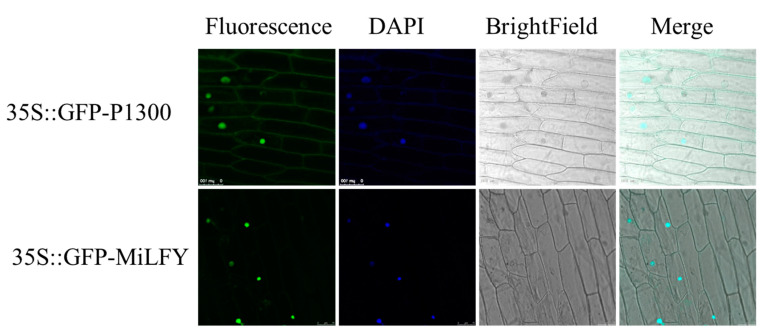
Subcellular localization analysis of the MiLFY protein. The subcellular localization of 35S::GFP-MiLFY and 35S::GFP-P1300 in onion epidermal cells was evaluated.

**Figure 4 ijms-23-03974-f004:**
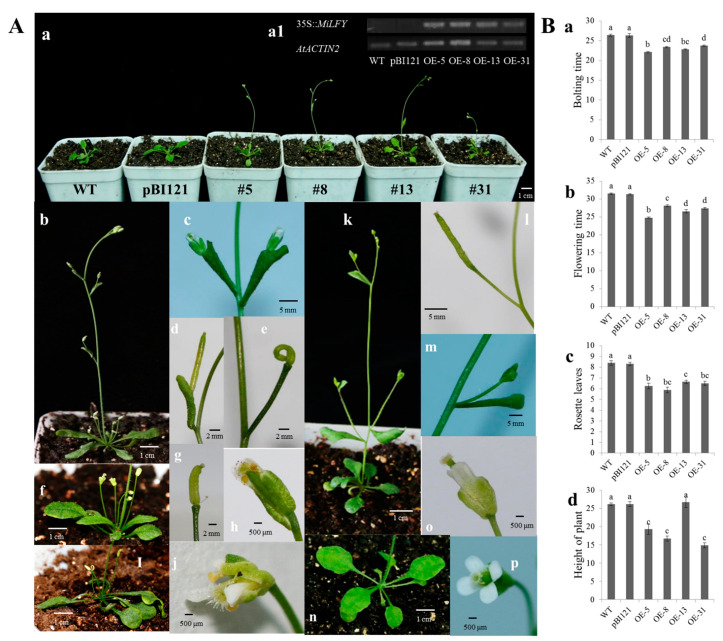
Phenotypic analysis of MiLFY-overexpressing transgenic plants. (**A**) (**a**) Phenotypes of *MiLFY*-overexpressing *Arabidopsis*, transgenic pBI121 empty vector *Arabidopsis*, and WT *Arabidopsis* at different flowering times. (**a****1**) A semiquantitative method was used to detect the expression of *MiLFY* in transgenic plants. *AtActin2* was expressed in both transgenic and control plants, and the original electrophoretic results are shown in Appendix A. (**b**–**j**) Specific phenotypes of MiLFY-overexpressing *Arabidopsis*: (**b**) plant phenotype, (**c**) opposite or whorled stem branches and curly leaves, (**d**) siliques and curly leaves, (**e**) slightly curled siliques, (**f**) single flowering of basal branches, (**g**,**i**) small and short siliques, and (**h**,**j**) flowers with few petals. (**k**–**p**) Normal phenotypes of WT *Arabidopsis*: (**k**) plant phenotype, (**l**) siliques, (**m**) stem branches, (**o**,**p**) flowers, and (**n**) basal branches. (**B**) (**a**) Comparison of bolting time between *MiLFY* transgenic plants and the control lines. Bolting time was measured when the plant was bolting until the height was approximately 1 cm. (**b**) Comparison of flowering time between *MiLFY* transgenic plants and the control lines. The flowering time was measured when the first flower opened. (**c**) Comparison of the number of rosette leaves between *MiLFY* transgenic plants and the control lines. The number of rosette leaves was determined when the plants were bolting. (**d**) Comparison of plant height between *MiLFY* transgenic plants and the control lines. The height of the plants was measured 15 d after flowering. Significant differences among the samples were assessed at the *p* < 0.01 level by Student’s t tests; different letters indicate significantly between different samples.

**Figure 5 ijms-23-03974-f005:**
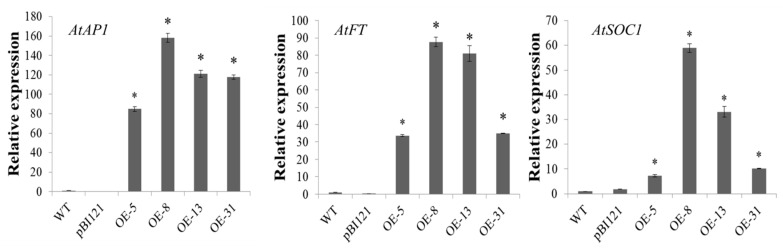
Expression of the flowering-related genes *AtAP1*, *AtFT*, and *AtSOC1* verified by qRT–PCR in transgenic and control lines. Significant differences among the samples were assessed at the *p* < 0.01 level by Student’s t tests; asterisks (*) indicate samples significantly different from the wild type.

**Figure 6 ijms-23-03974-f006:**
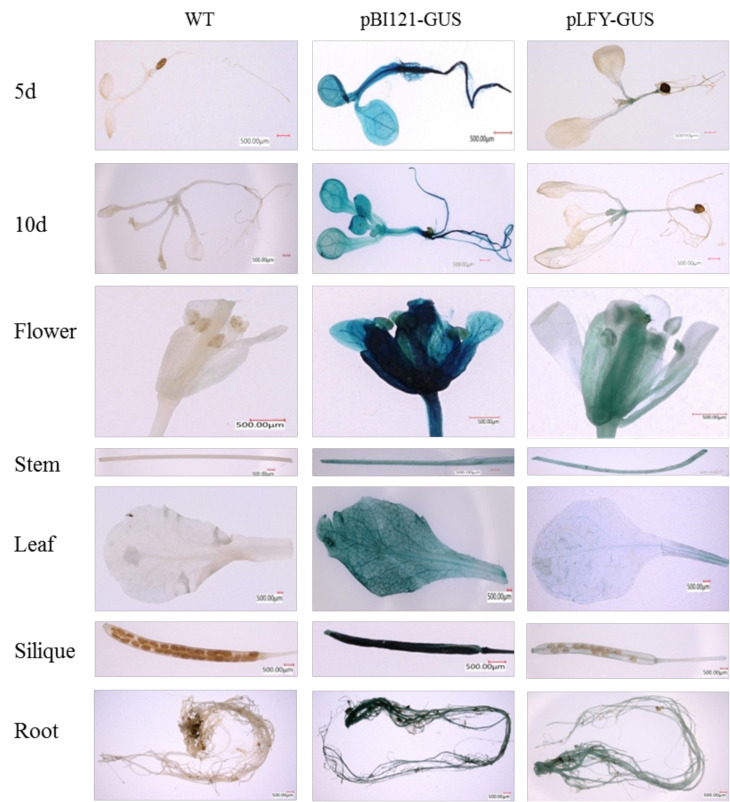
GUS staining in different organs and at different stages in transgenic pLFY-GUS plants, pBI121-GUS vector plants and WT plants.

**Figure 7 ijms-23-03974-f007:**
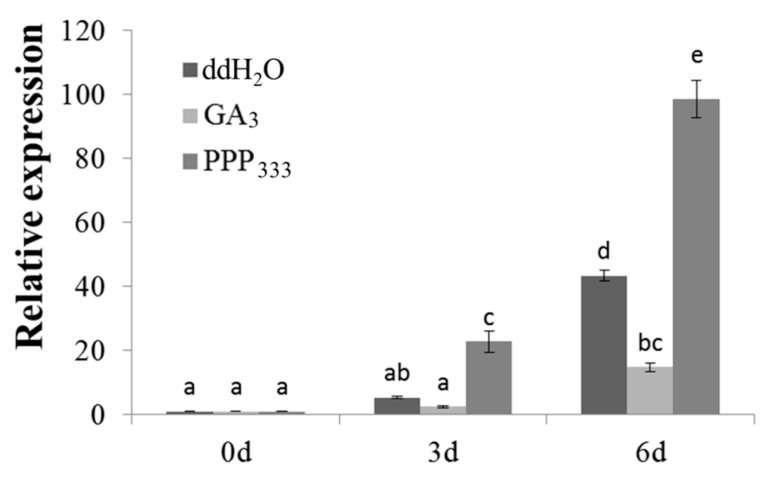
*GUS* expression in transgenic *Arabidopsis* plants treated with different PGRs treatment. Significant differences among the samples were assessed at the *p* < 0.01 level by Student’s t tests; different letters indicate significance between different samples.

**Figure 8 ijms-23-03974-f008:**
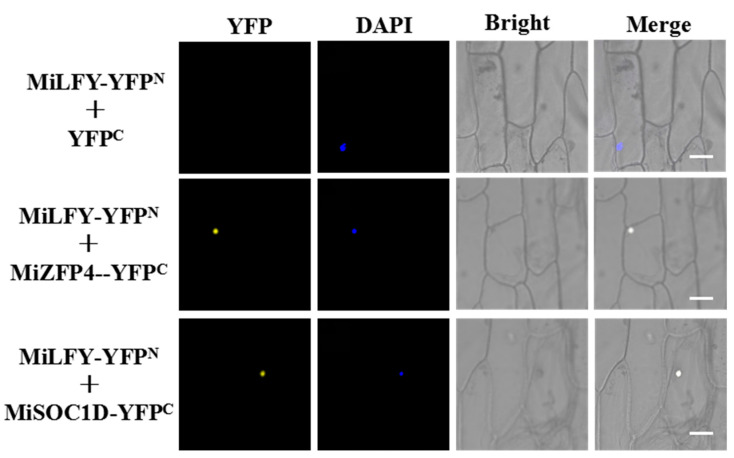
BiFC assays of MiLFY and candidate proteins. Bars = 20 μm.

## Data Availability

Not applicable.

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
