# Peer review of "Isolation and Functional Characterization of a LEAFY Gene in Mango (Mangifera indica L.)"

_ijms, 2022, doi:10.3390/ijms23073974_

Round 1
Reviewer 1 Report
The characterisation of the MANGO LFY gene is of interest both from the perspective of molecular understanding and the importance of flowering time of the crop for commercial production.
The manuscript is largely readable and would seem suitable for publication if several issues are addressed. It is difficult to review some parts of the manuscript in the current form due to inconsistencies e.g. the M&M and results sections relating to the subcellular localisation seem to be two different experiments!
The main issue is that the materials and methods lack detail and would not allow a reader to replicate the experiment. This added the lack of clarity in some of the results makes it impossible to review the results with confidence.
The best example is section 2.3 - Subcellular Localization of MiLFY.
The results state “A P1300 vector containing the MiLFY promoter fused to GFP and a P1300-GFP 119
empty vector (a control) were used in this study”. This suggests a MiLFY promoter fusion was used to drive GFP. If this were the case the (nuclear) localisation observed would be the result of GFP protein not the promoter.
No details are given of P1300 vector? A citation would be useful. The same is true of other vectors used such as pBI121. P1300-GFP empty vector control is mentioned. It is not clear what this is, but I assume it is not an empty vector as it has GFP expression is directed by some form of promoter.
This vector also leads to nuclear expression. Is this a result of a nuclear localisation signal on the GFP protein?
The materials and methods (section 4.5 - Subcellular Localization Analysis) seem to suggest an entirely different experiment?
These state, I assume correctly, that an MiLFY-GFP protein fusion not driven by the miLFY promoter?
The M&M are too scant to allow proper evaluation. Details are needed of the P1300 vector. There is not details for the control used, nature of fusion etc.
Why does GFP localise to the nucleus from the vector control? Would the GFP protein used in the fusion be enough to direct the protein here?
The materials and methods for the 4.4- Analysis of MiLFY Gene Expression also lack detail Same for GUS expression in Fig. 7).
For the 2-ΔΔCt method matching PCR efficiency, close to 100%, is required. No details of this are given.
The 2-ΔΔCt also requires a refence sample to which are comparisons are made. From the methods it is not clear which sample was used as reference.
Is relative expression a fold change difference compared to the reference or some other calculation.
If fold change, some of the changes are small and I would question the robustness of such small changes with so few replicates.
It is stated, in section 4.1, that all samples were collected from the same plant. Here (4.4) it is stated that at least 3 biological replicates were used. Are 3 tissue samples collected from the same plant at the same time biological replicates or technical replicates identifying difference is processing?
Fig 1. Would be improved if C were below A % B rather than alongside. Currently section B is so small it is difficult to read.
The English is generally of an acceptable quality but would benefit from an additional proofread.
Some minor changes are required, couple examples:
Ln15 (and Ln76) – encoded a protein of 383 amino acids.
Ln 76 ORF is given as 1152bp whereas in the abstract the “cDNA sequence of MiLFY
was 1149 bp. Be more consistent, is this really a cDNA length or the ORF minus the stop codon?
Ln 83 Longan DILFY needs the species name?
Ln 131 Is pBI121-Gus a transgenic control or a positive control?
Ln170 Is pBI121 empty vector control really an empty vector or does in express Gus?
Ln374 “dipped in GUS staining buffer” implies immersed for a short period whereas they were left for 24h.
Reviewer 2 Report
Dear Authors,
I have found your manuscript well written and mature. It provides significant information for Mango breeders, as well as for plant physiologists working on flowering process. I have only minor suggestions:
line 66 - please provide the aim of the study in a separate paragraph
Figure 2, 7 - description of Y-axis - shouldn't it be "relative expression"?
line 170 - 35S::MiLFY
the usage of "phytohormone" word - it is always safer to use plant growth regulator (PGR) instead, since "phytohormones" refers exclusively to plant produced PGRs. Paclobutrazol is synthetic, exclusively exogenously added chemical compound, thus it cannot be named as "phytohormone". Please, check lines for example 197, 200, 209.
line 280, 298 - PPP instead of PP
Line 379-378 - provide information on the way of application of PGRs, whether plants were sprayed or watered with the given solution, which volume was applied per plant?
Best regards and good luck with your experiments
Round 2
Reviewer 1 Report
This is the second time i have seen this manuscript and I find it to be much improved. There are however still some areas that must be improved.
As in the previous version, although the text mentions Fig. 5B there is no Fig. 5 in the manuscript and no results associated with the expression analysis described. Also although Fig. 5B is mentioned in the text 5A is not - is there a 5A?
The methods describing the quantitative real time PCR are still vague. The method used, as in the reference given in the manuscript, is dependent on PCR primer pairs having similar PCR efficiencies that are these are close to 100%. It is usual to give the PCR efficiencies to confirm the methodology is valid (and how these were calculated).
Other suggestions below:
Ln 19 – the fusion protein is localised to the nucleus, not expressed in the nucleus.
Ln 55 – please say what the NtNFL1 gene is?
Ln 58 – there are no references to support the comments in this paragraph.
Ln 83 – without referring the to figure 1 it is difficult to know which species are referred to with “DlLFY and CsLFY”
Ln 111 – in the figure legend explain the letters (a,b,c) etc rather than just (Duncan’s test: P < 0.01).
Ln 132 – states that “exogenous and endogenous expression levels in the transgenic plants were measured". This is a little confusing. If I understand correctly, two different methods were used to measure this. Is this correct? To me. this sentence suggests that exogenous and endogenous expression of LFY (Mi and At) where measured in Arabidopsis. I don’t think this is correct as there are no primers for AtLFY.
Ln 133 - and in other places it is mentioned that semi quantitative PCR was used to measure gene expression. There are no materials and methods for this - please add. The results for this (fig. Aa) are difficult to see but it appears to have unequal loading i.e. WT has faint actin (and slightly quicker migration) and no visible MiLFY whereas actin is higher in the OE samples.
Ln 135 - refers to Fig. 5B There is no figure 5B!!!!!
Ln 164 – suggests the qRT PCR data on Arabidopsis is in Fig. 5. Again, there is no Fig. 5. I cannot see this data anywhere?
Ln 167 - suggests that AP1, FT, and Soc are upregulated. What about AtLFY. Ln 132 states that endogenous expression (of AtLFY?) was measured but no mention and no data.
Ln 208 – In Fig. 7 legend please make it clear that the “GUS expression” is the transcript not GUS protein. It was only from the M&M that this was apparent. Also, in Supplementary Table 1, primers for GUS fluorometric assays are given. What are these used for?
Ln 337 - I am unfamiliar with this method: “To avoid large differences in the data between samples, the average ΔCt value of all samples was used as the reference to calculate fold expression.” I do not understand what you mean by this. What large differences are you referring to? Can you give a reference to support this analysis method as it does not seem to be supported by the reference given? i.e. taking an average of all samples rather than those of a nominated reference sample(s).
Ln 337 – was the Arabidopsis expression data derived in the same way?
Line 340 – the 2-ΔΔCT method is dependent on the PCR efficiencies of the primers used being similar and both close to 100%. Please supply information on the efficiency of the primers used for MiLFY Actin and the Arabidopsis genes tested.
Ln 345 – please give a reference for the P1300 vector.
Ln 367 – where can I find the primer sequences for the Arabidopsis actin gene and the genes analyzed in Arabidopsis? If Supplementary Fig. 1 please give this.
